# Lifestyle Habits and Health Indicators in Migrants and Native Schoolchildren in Chile

**DOI:** 10.3390/ijerph18115855

**Published:** 2021-05-29

**Authors:** Mónica Suárez-Reyes, Daiana Quintiliano-Scarpelli, Anna Pinheiro Fernandes, Cristian Cofré-Bolados, Tito Pizarro

**Affiliations:** 1Escuela de Ciencias de la Actividad Física, el Deporte y la Salud, Universidad de Santiago de Chile, Santiago 9170020, Chile; monica.suarez@usach.cl (M.S.-R.); cristian.cofre@usach.cl (C.C.-B.); 2Carrera de Nutrición y Dietética, Facultad de Medicina-Clínica Alemana, Universidad del Desarrollo, Santiago 7610658, Chile; apinheiro@udd.cl; 3Facultad de Ciencias Médicas, Universidad de Santiago de Chile, Santiago 9170020, Chile; tito.pizarro@usach.cl

**Keywords:** nutritional status, foreigners, schoolchildren, physical condition, diet

## Abstract

In Chile, the migrant population has increased in the last years. Migrants adopt behaviors of this new culture, which can have an effect on their health. Contradictory results regarding differences between migrant and native children have been reported. The aim of this study was to explore the associations between nationality with health indicator and lifestyle habits among schoolchildren in Chile. A cross-sectional and observational study with a non-probabilistic sample was conducted in 1033 children (86.4% Chilean and 13.6% migrant) from second to fourth grade of seven public schools from low-income municipalities from the Metropolitan Region, Chile. Anthropometric measurements (weight, height, waist circumference, triceps and subscapular folds), handgrip strength, and standing long jump measurements, physical activity, self-esteem and food guidelines accomplishments were determined. Migrant children presented lower body mass index (BMI), Z-BMI, body fat% and waist circumference values; and higher handgrip strength, standing long jump, and more satisfactory compliance with food guidelines than natives (*p* < 0.05). No significant difference for physical activity and self-esteem was observed. In the adjusted models, migrants presented lower odds for overweight/obesity, risk of abdominal obesity, low handgrip strength and unsatisfactory food guidelines accomplishment in all models (*p* < 0.05). The nutritional and muscular fitness of migrant children was better than that of the Chilean ones.

## 1. Introduction

Migration refers to the movement of people from one territory to another, temporarily or permanently, to usually settle to live [1]. This is a phenomenon that has been increasing continuously in recent years [2]. People who migrate acquire the status of migrant which has been associated with various factors that affect people’s health. Among the factors that affect health are the changes in habits and behaviors that are required to adapt to a new culture and assimilate the new context [3]. The assimilation refers to the process related to the interaction between two cultures regarding social norms beliefs and behaviors [4]. Evidence reports that there is a greater degree of assimilation in generations born in the host country. In cases that are the first generation to migrate, assimilation will depend on the age of arrival and time of stay. In countries with a longer history of immigration, it has been reported that being a migrant is associated with worse eating habits, a higher risk of overweight and chronic diseases [5,6]. This can have adverse effects on current and future health, especially in children. Children who migrate experience a change in the environment in which they develop. Cultural differences can influence children’s habits and consequently have an effect on their physical and mental development [4].

It has been described that when migrants adopt the diet of the new culture, they gradually lose that of their culture of origin, which is linked to many factors, being the availability of typical foods in the new culture one of the most important [7]. On the other hand, studies on migrants of Latino origin, has been reported that those with less assimilation are more protected against health problems, and that those with low income have a lower rates of healthy eating, given by lower consumption of fruits and vegetables and higher consumption of processed foods [8,9]. 

Proper eating and physical activity habits are important for the physical and mental development of children. Studies have reported contradictory results regarding differences between migrant and native children. A study in Spain reported that migrant children have a lower prevalence of obesity, higher levels of physical activity and better physical condition [10]. On the other hand, children with an immigrant background may have a higher risk of low levels of physical activity and a higher risk of overweight, but better nutritional recommendations accomplishment [11].

The assimilation process can be stressful. In addition to having effects on a physical level, it can also have effects on a mental level. A study carried out in Chile reported that there are no differences in mental health indicators between migrant and native children. However, the study shows that migrant children have fewer personal resources such as self-esteem or social integration [12]. This could have negative effects on integration into the host society. Schools play an important role mediating the health disparity between migrant and non-migrant children. In the school setting, information can be transmitted through children and to the family by addressing cultural barriers and avoiding potentially stigmatizing approaches, especially related to health messages [13].

In Chile, the increase in the migrant population is due to the country’s favorable economic situation and job stability compared to other Latin American nations. On the other hand, Chile has low global diet quality indicators and one of the highest rates of overweight/obesity in the region [14,15]. According to national statistics, the migrant population amounts to 1,492,533 of people [16]. Most of the people who migrate to Chile come from countries of the Latin American region, especially from Venezuela, Peru, Colombia, and Haiti [17]. Part of the population that migrates to Chile consists of school-age children (10.6% are under 14 y) [18]. This new population of schoolchildren is registered mainly in the public education system, representing 4.4% of total enrollment in the school system in the country [19]. Despite the fact that more and more migrant children are in Chile, and that a national policy has recently been developed, there is little information on their characteristics and their integration into the school system [20]. Thus, our goal was to explore the associations between nationality (migrant and native) with health indicators and lifestyle habits among schoolchildren in Chile, considering differences by sex, in 6 municipalities of the Metropolitan Region.

## 2. Materials and Methods 

### 2.1. Design, Setting and Subjects

This is a secondary analysis within the framework of a cross-sectional observational study called “Development, scaling up and validation of an integrated system of interventions in schoolchildren in nutrition, physical activity and community environment”. From a non-probabilistic sample, seven public schools from six low-income municipalities of Santiago, Chile (El Bosque, La Granja, San Ramón, Lo Espejo, San Joaquín, and Pedro Aguirre Cerda) were invited to participate and compose the setting. In Chile, the proportion of children living in a vulnerable situation who attend public schools is reported by an index associated with poverty [21]. This index has a scale from 0 to 100, where the greater the proportion of children in situations of vulnerability due to poverty, the greater the index. In our sample, the average index of participating schools was 91.5 ± 5.4 (min: 88.0; max: 98.0), which reflects high vulnerability due to poverty. The studied population consisted of students from second to fourth grade of primary education, who in general range from 7 to 10 years of age. Children were included if they meet the following criteria: (i) not having any disease or physical problem limiting their capability to perform any of the tests or measurements, (ii) not having any cognitive condition limiting their capability to answer any of the questionnaires. We excluded children with missing data of height (*n* = 5), handgrip strength (*n* = 2), standing long jump (*n* = 3), self-esteem (*n* = 15), physical activity (*n* = 6), skinfolds (*n* = 2), waist circumference (*n* = 2), as well as those with implausible values (*n* = 2). Since we did not have information on sexual maturation stages, children equal or older than 12 years old were also excluded from this analysis (*n* = 15). In sum, fifty-two children were excluded, and the final sample included 1033 children (7 to 11.9 years old).

### 2.2. Instruments and Data Collection 

Data were collected by staff trained from October to December 2019, and this protocol was approved by the Ethics Committee of the Universidad de Santiago de Chile (record number 187/2019). Sociodemographic variables considered were sex, age, and nationality. The variable time of migration was used to adjust multivariable models (reference category less than 1 year, followed by 1–2; 3–5; 5–8, and more than 8 years). All anthropometric measurements (weight, height, waist circumference, triceps, and subscapular folds) followed the National Health and Nutrition Examination Survey [22], using an inelastic tape for circumferences; scale Seca 813 (Seca, Hamburg, Germany), for weight; stadiometer Seca 213 (Seca, Hamburg, Germany) for height and Lange^®^ adipometer (Bloomington, Minnesota, USA) for skinfolds. Nutritional status was classified according to Z-BMI/Age, calculated using Anthro Plus (WHO) software. Abdominal obesity by percentile of Fernández [23], classified as normal if values of waist of circumference were <p75; risk of abdominal obesity between p75–p90; and abdominal obesity if *p* ≥ 90. Body fat percentage was calculated by Slaughter equations [24].

Handgrip strength was estimated with a Jamar^®^ PC-5030 hand hydraulic dynamometer (Jamar Dynamometer, Lafayette, IN, USA); children performed two attempts with each hand. The average between the maximum values (kg) for each hand (left and right) was calculated and considered for analysis. The standing long jump test was used to measure explosive power of the legs. In this test, children jumped as far as they could with their feet together. The distance between toes at take-off and heels at landing were recorded to the nearest 0.1 cm. Children attempted three times and the best score was used for analysis. Both tests have been used in children and have high reliability [25]. Due to the lack of consensus on reference values for these tests [26], we defined at risk of low muscle strength as *p* < 25 by age and sex for each test (Table 1). 

Self-esteem was evaluated with the Rosenberg test, which consists of ten questions rated from 1 to 4 (highly disagree to highly agree) and negative items are reverse scored. The score can be between 10 and 40 points, where a higher score indicates higher self-esteem. Children’s self-esteem was classified as high (≥30 points), medium (26–29 points), or low (≤25 points) [27].

Physical activity was estimated using the Physical Activity Questionnaire for Older Children (PAQ-C). Through this questionnaire, it is possible to estimate the moderate to vigorous physical activity that children engaged during the past 7 days. The PAQ-C consists of nine items rated from 1 to 5 and higher score indicates that children are more active [28].

To assess the dietary habits, five questions based on the Chilean Food Guidelines for children over 2 years were considered [29]. The answers to each question were categorized according to the food intake recommendation: daily water intake (<6 glasses; ≥6 glasses); weekly frequency of legume consumption (beans, lentils, peas, chickpeas) (<2 times; ≥2 times); daily frequency of dairy intake (<3 times; ≥3 times); weekly frequency of fish and seafood (<2 times; ≥2 times); and daily portions of fruits and vegetables (<5 portions; ≥5 portions). Based on the Chilean National Food Consumption Survey (15), a global index of food intake compliance was created, and grouped the children who met the recommendations according to the food guidelines. For this purpose, “satisfactory compliance” was considered if compliance with at least three of the five guidelines was reported; compliance with one or two guidelines was considered “partial compliance;” and “non-compliance,” when none of the five guidelines was reported.

### 2.3. Data Analysis

Descriptive data of categorical variables were expressed in relative and absolute frequencies. Numeric data were described as medians, interquartile ranges according to sex and nationality. The Kolmogorov–Smirnov test was used to determine the normality of the distribution. For bivariate analysis assessing the sex differences in covariates, the Chi-square, Kruskal–Wallis and Mann–Whitney U tests were used, depending on variable type. Logistic regression was used to assess the association between anthropometric status, physical condition, self-esteem, and food guidelines compliance, stratified by sex and nationality. Multivariable models were adjusted for the following covariates: time of residence in Chile, PAQ-C final score, and Z-BMI/A and age regardless of their level of significance. This is based on the available evidence the aforementioned variables have been associated with the outcomes of our study [3,7,30,31]. Goodness of fit was measured with the Hosmer–Lemeshow test. Alpha was set at *p* < 0.05. Statistical analysis was carried out using STATA 16.1 for Mac (StataCorp LLC, College Station, TX, USA).

## 3. Results

Of the total sample (*n* = 1033), 86.4% were Chilean and 13.6% were migrant children. Considering the total number of migrant children, 39.3% came from Haiti; 22.1%, from Venezuela; 12.9%, from Peru; 10.7%, from the Dominican Republic; while the remaining 15.0% included children from Colombia, Bolivia, Argentina, and Mexico. Most of the migrant children (68.2%) had up to 1 year of migration to Chile. Table 2 shows the general anthropometric and physical characteristics of the sample, by sex and nationality. Differences according to nationality were observed in almost all variables except for age, physical activity, and self-esteem. Migrant boys were thinner, while migrant girls were taller than their Chilean peers. Consequently, migrants of both sexes had lower BMI and Z-BMI values.

Considering the whole sample, according to body fat and its distribution, migrant children showed 7.7% lower body fat and almost 6 cm less of waist circumference. Migrant children also obtained 1.0 kg more of absolute values of handgrip strength and reached 8.0 cm more in the standing long jump test. The same trends were observed in relative value analysis and by sex stratification (*p* < 0.05).

Table 3 describes the association of categorical variables in relation to sex and nationality. For nutritional status and abdominal obesity, considering general and comparison between Chilean girls and boys compared to their migrant peers, a lower proportion of excessive weight (−25%) and abdominal obesity (−18.5%) in migrants (*p* < 0.001) was observed. No statistical difference was observed for self-esteem, the majority being classified as medium (about 40% of the sample). In relation to physical condition variables (absolute and relative handgrip strength; absolute and relative standing long jump), a higher proportion of those classified as *p* < 25 values were observed in Chilean children. For absolute and relative handgrip strength, significant differences were observed for general (*p* = 0.007; *p* = 0.001, respectively) and for boys (*p* = 0.022; *p* = 0.001, respectively). Considering the relative standing long jump, a significant difference was observed just for the general sample, with a higher proportion of Chilean children with low performance *p* < 25 compared to the migrants (*p* = 0.013).

Finally, in relation to the Food Dietary Guidelines compliance, the less unaccomplished was the water intake (6.6%), with no difference by nationality and sex. The most accomplished was the fruit and vegetables intake, mostly due to fruit consumption, with a general proportion of 44.0%; with significantly higher totals in migrants and of these, girls, compared to Chilean (*p* = 0.033; *p* = 0.009, respectively). Satisfactory accomplishment was observed in less than one quarter of the studied population, being higher in migrants (*p* = 0.006), especially in girls (*p* = 0.012) (Table 3).

Figure 1 describes the not adjusted and adjusted models for risk of obesity/overweight, abdominal obesity, low handgrip strength and standing long jump, low self-esteem, and unsatisfactory accomplishment of food guidelines in Chilean and migrant children general and stratified by sex. For overweight/obesity and risk of abdominal obesity, significantly lower odds were observed for migrants in all models (model 1: 0.35 [0.24–0.52]; model 2: 0.36 [0.25–0.53]; and model 3: 0.23 [0.13–0.41]; (*p* < 0.05). Adjusted by time of migration, these odds lowered by 36.1% and 38.5%, respectively, where the time of migration was a factor to control weight gain (*p* < 0.001; data not shown).

In relation to low handgrip strength (<p25) lower odds in the general sample for migrants in all models were observed when compared to Chileans (*p* < 0.05). When stratified by sex, this trend remained significant in Model 2 only for boys, but when adjusted by time of migration (Model 3), significance was lost. Low standing long jump (<p25) was significant just in the univariate model for the general sample (OR: 0.55; CI: 0.35–0.89). Low self-esteem was not significantly associated to nationality or sex in any model. Unsatisfactory food guidelines accomplishment presented lower odds in migrants for the general sample (Models 1 and 2). In the analyses by sex significance is only observed in girls (Models 1 and 2).

## 4. Discussion

In our sample migrant schoolchildren presented better health indicators and lifestyle habits than the native ones, except for self-esteem that was similar in the groups. Some of these differences were possible to evidence just with the sex stratification (risk of low handgrip strength, for boys; and risk of unsatisfactory/partial food guidelines accomplishment, for girls). It is known that Chile leads the rates of malnutrition due to excess in adults and children in the Latin–American region [14]. However, the impact of this reality in migrant children is still unknown. We observed a prevalence of overweight/obesity of 56.9% in Chilean children and 32.1% in migrants. The prevalence observed in our study among Chilean children is slightly higher than previously reported (51.3%), while it is lower in migrant children (38.0% previous data) [32].

It has been described that in some cases when children arrive for the first time to a host country, they are healthier than their native peers. This phenomenon is known as the “healthy immigrant effect” [33,34]. Compared to compatriots who do not emigrate, those who do are healthier, have less obesity, and better life habits [33], which is mainly observed in people who migrate from a developing country to a developed country. This could be the case in our study, where those who arrive in Chile, and compose our sample, usually come from lower income countries [35]. Depending on the length of the stay, this effect can be altered by the degree of assimilation, the resources of the culture of origin, and the level of integration to the local culture [36]. Thus, the health and nutritional status of migrants may worsen as they adopt Chilean habits related to diet and physical activity.

Mora et al. [37] showed the short-term impact of migration processes on nutritional status and eating habits of Moroccan adolescents living in Spain. They found a higher prevalence of overweight and obesity in those migrants living in Madrid, compared to native Spanish ones of the same age. Another study determined the nutritional status in children from low-income countries who migrated to Spain, and found that BMI increased by 0.02 for each month of stay in Spain [38]. Considering this data, each year in the host country would relate to an increase of 0.24 kg/m^2^. At this moment, no national longitudinal data could prove if this phenomenon would occur in our migrants.

Few investigations have focused on muscular fitness in migrant children. We observed better performance in both tests of handgrip and standing long jump in migrants compared to Chilean children. In addition, regarding the risk of low performance, the results were significant only for handgrip strength in all models, and mainly in boys. Our results agree only partially with results reported by Pardo-Arquero et al. [10] who did not observe differences for handgrip strength between migrant and native Spanish children, either in boys or girls. Otherwise, they reported that migrant girls reached greater distances than native Spanish girls in the jumping test.

Findings reported by Pardo-Arquero et al. [10] are not consistent with other studies where physical fitness of migrant children has been studied. Thus, in Switzerland, children from migrant families had lower muscular fitness related to less physical activity levels than their native peers. The lower physical activity level of migrant children was because migrant children are less likely to participate in programmed physical activity (sports) than natives [39]. Physical activity is one of the determinants of physical fitness including muscular fitness. Thus, differences in levels of physical activity could explain the performance of children in different tests. However, we only observed differences in the levels of physical activity in boys. Physical activity has been more studied in migrant children, although the results are contradictory. Higher levels of physical activity in the migrant population have been previously reported in other contexts [34,40]. On the other hand, Besharat et al. [11] reported that children with a history of immigration in Sweden have a higher risk of low physical activity compared to natives. The use of different types of measurements and methodologies could explain the discordant results.

It is worth mentioning that the type of physical activity, as well as its intensity, must be adequate to have an effect on physical and muscular fitness. Thus, lower participation of migrant children in programmed physical sports activities has been reported [10,40]. This could be due to their families’ lack of information and resources. The migrant children our study, especially boys, could have adapted better to available resources to be more active, their physical condition may be more determined by genetics, or family habits many have played a protective role. Further studies are needed to elucidate such assumptions.

In the present study, no differences were observed with regard to self-esteem or the risk of low self-esteem by nationality or sex. Our results do not agree with the results of Caqueo et al. [12], who reported lower self-esteem scores in migrant children compared to Chilean primary school children. Caqueo et al., did not observe differences in other mental health indicators, and differences between Chileans and immigrants regarding self-esteem were not observed in secondary school children and adolescents. The fact that the immigrant children in our sample have similar self-esteem to the Chilean children represents a personal resource that would allow them to adapt psychologically to changes related to the migration and assimilation process. This adaptation resource has previously been defined as a phenomenon called “the immigrant paradox” [41]. Low self-esteem in migrant children is associated with the level of stress or trauma linked to the migration process. Therefore, it would be more common in children who migrate for humanitarian reasons, which would not be the case for the majority of immigrants who arrive in Chile.

With regard to dietary habits in the general sample, the migrant children presented better habits than the native ones, but the food guidelines accomplishment is insufficient in both groups, since the satisfactory accomplishment proportion was around 20% for Chileans and 30% for migrants, where migrant girls had the highest rate of accomplishment. Further, fish and legume consumption were higher in migrants than in Chileans; this could be due to the Caribbean countries of origin in the sample. Evidence usually reports worse habits in migrant populations [42,43], but considering that the host territory has direct repercussions on the diet consumption, is expected that, in time, the habits of those who arrive in Chile will become worse, since our country has relatively low global diet quality indicators and, as mentioned before, the highest rates of excess weight of the Region [14,15].

A similar tendency was observed in Australia, considering a cohort analysis of adolescents. The data showed that the intake of foods high in saturated or trans fatty acids was positively associated with length of stay in the country. As time progressed, dietary intake of migrant children changed to negative with possible consequences to their health [44]. In China, migrant students exhibited lower percentages in their daily intake of vegetables (57.3 vs. 63.7%), fruits (27.7 vs. 38.3%), meats (37.0 vs. 44.3%), soybean (11.6 vs. 17.5%) and dairy products (28.4 vs. 42.5%) than native children [45]. Finally, according to Besharat et al. [11] offspring of immigrants complied more fully with nutritional recommendations but had a higher risk of having low physical activity and being overweight, compared to children of Swedes. Pávez and Durán [46] discussed that migrant children can imitate the eating habits of local children, characterized by the consumption of foods and beverages with a high caloric content, fat, sugar and salt, because during the migratory process, children’s eating habits are transformed as a way of cultural adaptation. In Spain, in immigrants of different origins, worse compliance with the recommendations than in natives was observed. However, considering their origin and cultural and geographical characteristics, natives and immigrants from Mediterranean countries showed healthier eating patterns than immigrants from non-Mediterranean countries. However, the quality of diet was worse than their peers in the country of origin, except in the case of non-Mediterranean immigrants, whose quality of diet improves when residing in Spain [47]. Dietary habits and physical activity behaviors are influenced by post-migration environments, culture, religion, and food or physical activity-related beliefs and perceptions and availability [48]. So further studies are required to understand the influence this and other migration-related factors have on behavior changes after migration.

In response to the growing migrant population and the challenges that this entails, in the last 5 years Chile has developed policies to protect the health and access to education and associated benefits of those who arrive in the country [17,49]. The implementation of these public policies is a great challenge. Since, for a real effectiveness it requires the recognition of cultural conceptions related to health (for example, self-care, nutrition, parenting guidelines) and strengthening intercultural relationships in educational communities. In a way, programs that promote the exchange of knowledge related to healthy habits, where migrants and natives can benefit for a better quality of health and life, should be our greatest legacy for future migrants and native children in Chile.

Our results must be interpreted considering some limitations. First, the original study from which this secondary analysis was made was not designed for the purpose of this study. The use of a non-probabilistic, non-representative and convenient sample does not allow these findings to be extrapolated to all migrant children who have arrived in Chile. Despite this, in our results we observed associations between nationality and indicators of health and life habits. Second, the cross-sectional design of the study does not allow for observing the evolution of the studied aspects. The length of the stay should be considered a factor that influences health-related habits of the migrant population. In the same way, it is important to point out the need to study contextual variables such as parents’ socioeconomic and educational background. This, because the migration process involves, to a greater or lesser degree, adapting to a new environment, which could depend on the ability to access and use the available resources.

There is little information on the health status of migrant children in Chile. This study highlights the need to consider specific health-related lifestyle programs to protect this population of the obesogenic environment, considering the pertinence, and cultural barriers which should be addressed. This would be useful to promote habits that will positively influence healthy development in those at risk for health alterations.

## 5. Conclusions

In summary, in our sample, migrant children have better health indicators than their native peers in Chile. Migrant children have less prevalence of malnutrition due to excess and abdominal obesity. Also, migrant children, mainly boys, have better muscular fitness. Physical activity levels were not different, so other factors could affect differences we observed for muscular fitness. Self-esteem was similar in both groups, so this would represent a resource for the assimilation process of migrant children. Chilean food guidelines accomplishment was higher in migrants, especially the girls.

Finally, studies of this type should be furthered by other follow-up studies to observe the evolution of the health status of migrant children to better understand how the assimilation process may affect their development. Additionally, as a pioneer study in our country, our findings should remain been explored considering that this condition was found in sectors with high social vulnerability and could help the public health decision makers to identify practices that can be implemented in educational communities, since migration follows growing in Chile.

## Figures and Tables

**Figure 1 ijerph-18-05855-f001:**
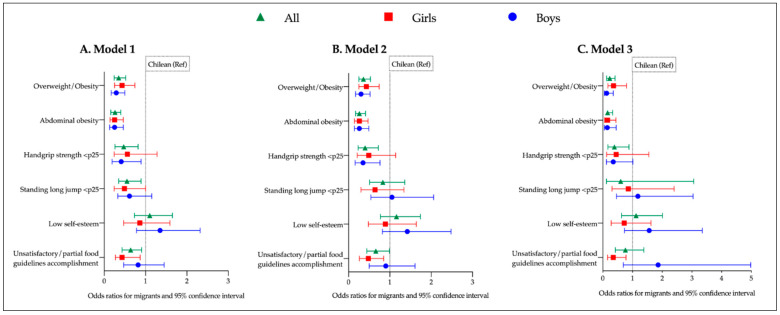
Associations models between nationality with anthropometric variables, physical condition, self-esteem and food guidelines accomplishment in all sample and by sex. (**A**). Model 1: not adjusted; (**B**). Model 2: all variables adjusted by final PAQ-C score. Handgrip strength and standing long jump additionally adjusted by Z-BMI. Low self-esteem and unsatisfactory/partial food guidelines accomplishment additionally adjusted by age. (**C**). Model 3, all variables adjusted by the same covariables as model 2 plus time of migration.

**Table 1 ijerph-18-05855-t001:** Cutoffs values to determine risk of low handgrip strength and standing long jump by sex and age.

Variables	Age				
	7.0–7.9	8.0–8.9	9.0–9.9	10.0–10.9	11.0–11.9
*n* girls/boys	96/108	135/147	134/165	85/119	13/31
Handgrip strength at p25 (kg)					
Girls	6.0	8.0	10.0	11.0	10.0
Boys	8.0	9.0	11.0	12.0	14.0
Relative handgrip to weight at p25 (kg/kg)					
Girls	0.25	0.27	0.27	0.29	0.23
Boys	0.29	0.27	0.30	0.28	0.29
Standing long jump at p25 (cm)					
Girls	80.5	84.3	87.5	91.0	81.0
Boys	88.4	94.0	97.1	98.2	110.6
Relative standing long jump at p25 to height (cm/cm)					
Girls	0.65	0.66	0.64	0.64	0.53
Boys	0.69	0.72	0.73	0.60	0.76

p25: percentile 25; kg: kilograms; cm: centimeters.

**Table 2 ijerph-18-05855-t002:** Descriptive statistics of continuous variables of the sample according to nationality and sex.

Variables	All		Girls		Boys	
	Chileans	Migrants	Chileans	Migrants	Chileans	Migrants
*n*	893	140	396	67	497	73
Age (years)	9.1 (8.2–9.9)	9.1 (8.1–10.1)	8.9 (8.1–9.8)	9.1 (8.3–10.1)	9.1(8.2–10.0)	9.0 (8.0–10.0)
Weight (kg)	34.0 (27.8–41.0)	31.0 (26.7–37.3)**	34.0 (27.7–41.0)	32.3 (27.1–39.7)	34.0 (28.0–41.3)	29.3 (26.1–34.9)**
Height (cm)	132.6 (127.1–139.2)	133.3 (128.5–141.5)	132.2 (126.6–139.0)	134.0 (130.1–142.0)*	133.1 (127.8–139.5)	131.7 (128.1–139.6)
BMI (kg/m^2^)	18.8 (16.8–21.9)	17.2 (15.7–18.9)**	18.9 (16.9–21.6)	17.6 (16.0–19.9)*	18.6 (16.7–22.3)	16.6 (15.7–18.1)**
Z-score BMI/Age	1.25 (0.38–2.20)	0.53 (−0.25–1.21)**	1.25 (0.43–2.01)	0.75 (−0.18–1.17)**	1.22 (0.31–2.47)	0.25 (−0.44–1.26)**
Body fat (%)	29.1 (20.1–34.6)	21.4 (14.8–31.0)**	30.6 (24.6–35.4)	26.0 (19.3–34.6)**	25.4 (17.6–33.7)	16.4 (12.8–24.8)**
Waist (cm)	65.3 (59.8–73.6)	59.6 (55.6–64.5)**	66.1 (60.0–72.6)	60.6 (56.9–66.8)**	64.6 (59.3–74.5)	58.5 (55.1–63.2)**
Handgrip strength (kg)	12.0 (9.0–14.0)	13.0 (11.0–16.0)**	11.0 (8.0–14.0)	12.0 (10.0–16.0)**	12.0 (10.0–14.0)	13.0 (11.0–16.0)**
Standing long jump (cm)	102.8 (90.6–115.0)	110.5 (96.0–123.5)**	97.0 (84.1–107.0)	110.0 (92.0–118.7)**	107.0 (95.7–121.6)	114.8 (100.0–135.0)**
Relative handgrip strength (kg/kg)	0.33 (0.27–0.40)	0.41 (0.34–0.50)**	0.31 (0.25–0.37)	0.39 (0.29–0.49)**	0.34 (0.28–0.41)	0.43 (0.35–0.50)**
Relative standing long jump (cm/cm)	0.77 (0.67–0.87)	0.83 (0.72–0.95)**	0.74 (0.64–0.80)	0.81 (0.67–0.88)**	0.81 (0.70–0.91)	0.85 (0.75–0.99)**
PAQ-C	2.80 (2.42–3.23)	2.90 (2.51–3.30)	2.75 (2.40–3.14)	2.70 (2.41–3.27)	2.82 (2.44–3.29)	3.02 (2.63–3.39)*
Self-esteem score	28.0 (25.0–30.0)	28.0 (25.0–31.0)	28.0 (25.0–30.5)	28. 0 (26.0–31.0)	28.0 (26.0–30.0)	28.0 (25.0–30.0)

Data are expressed in median and interquartile range. Comparison of nationality and sex tested by Test U-Mann Whitney or *t*-Test by type of distribution. * *p* < 0.05; ** *p* < 0.01; *n*: number; BMI: body mass index; PAQ-C: Physical Activity Questionnaire for Children.

**Table 3 ijerph-18-05855-t003:** Descriptive statistics of categorical variables of the sample according to nationality and sex.

Variables	All		Girls		Boys	
	Chileans	Migrants	Chileans	Migrants	Chileans	Migrants
*n*	893	140	396	67	497	73
Nutritional status (%)	χ^2^ = 36.4; *p* < 0.001		χ^2^ = 11.0; *p* = 0.008		χ^2^ = 25.2; *p* < 0.001	
Underweight	3.3	7.9	2.5	6.0	3.8	9.6
Normal weight	39.9	60.0	39.7	56.7	40.0	63.0
Overweight	26.3	21.4	31.3	25.4	22.3	17.8
Obesity	30.6	10.7	26.5	11.9	33.8	9.6
Abdominal obesity (%)	χ^2^ = 42.5; *p* < 0.001		χ^2^ = 22.0; *p* < 0.001		χ^2^ = 23.1; *p* < 0.001	
Normal	49.9	79.3	45.2	76.1	53.7	82.2
Risk	22.3	11.4	27.0	10.5	18.5	12.3
Obesity	27.8	9.3	27.8	13.4	27.8	5.9
Self-esteem score classification (%)	χ^2^ = 0.38; *p* = 0.827		χ^2^ = 0.21; *p* = 0.900		χ^2^ = 1.4; *p* = 0.502	
Low	25.2	27.1	26.5	23.9	24.1	30.1
Medium	42.6	40.0	37.2	38.8	46.9	41.1
High	32.3	32.9	36.4	37.3	29.0	28.8
Handgrip strength < p25 (%)	χ^2^ = 7.2; *p* = 0.007		χ^2^ = 1.9; *p* = 0.167		χ^2^ = 5.3; *p* = 0.022	
	20.3	10.7	17.2	10.4	22.7	11.0
Standing long jump < p25 (%)	χ^2^ = 2.9; *p* = 0.089		χ^2^ = 1.3; *p* = 0.248		χ^2^ = 1.6; *p* = 0.212	
	26.0	19.3	26.3	14.9	26.2	17.8
Relative handgrip strength < p25 (%)	χ^2^ = 11.8; *p* = 0.001		χ^2^ = 2.4; *p* = 0.120		χ^2^ = 10.9; *p* = 0.001	
	28.0	14.3	28.5	19.4	27.6	9.6
Relative standing long jump < p25 (%)	χ^2^ = 6.2; *p* = 0.013		χ^2^ = 4.0; *p* = 0.046		χ^2^ = 2.4; *p* = 0.124	
	26.2	16.4	26.0	19.4	26.0	19.2
Adequate water intake ≥ 1.5 l/day (%)	χ^2^ = 0.66; *p* = 0.417		χ^2^ = 0.13; *p* = 0.724		χ^2^ = 2.1; *p* = 0.150	
	6.8	5.0	6.3	7.5	7.2	2.7
FV consumption ≥ 5 servings/day	χ^2^ = 4.5; *p* = 0.033		χ^2^ = 6.88; *p* = 0.009		χ^2^ = 0.25; *p* = 0.614	
%	42.7	52.8	38.7	56.7	45.9	49.2
Fish consumption ≥ 2 times/week	χ^2^ = 22.8; *p* < 0.001		χ^2^ = 15.2; *p* < 0.001		χ^2^ = 8.9; *p* = 0.003	
%	26.7	46.4	23.5	46.3	29.1	46.6
Legumes consumption ≥ 2 times/week	χ^2^ = 5.8; *p* = 0.016		χ^2^ = 5.2; *p* = 0.023		χ^2^ = 1.2; *p* = 0.277	
%	34.5	45.0	36.1	50.8	33.3	39.7
Dairy consumption ≥ 3 servings/day	χ^2^ = 0.65; *p* = 0.421		χ^2^ = 1.1; *p* = 0.303		χ^2^ = 0.03; *p* = 0.869	
%	40.0	43.6	38.1	44.8	41.4	42.5
General food guidelines accomplishment (%)	χ^2^ = 10.2; *p* = 0.006		χ^2^ = 8.8; *p* = 0.012		χ^2^ = 3.3; *p* = 0.189	
Unsatisfactory	24.6	13.6	26.5	13.4	23.1	13.7
Partial	54.2	57.1	54.0	53.7	54.3	60.1
Satisfactory	21.2	29.3	19.4	32.8	22.5	26.0

FV: fruits and vegetables; p25 cutoff point of the own sample according to sex and age.

## Data Availability

Not applicable.

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
