# Peer review of "Lifestyle Habits and Health Indicators in Migrants and Native Schoolchildren in Chile"

_ijerph, 2021, doi:10.3390/ijerph18115855_

Round 1

Reviewer 1 Report

Thank you for the opportunity to review this article. In my opinion it is a very well-written article.  I appreciate the researchers effort and novelty of this study. I have only minor comments:

This paper analyze the associations between lifestyle habits and health indicators in migrants and native children in Chile. Since this is a public healthy journal, I want to see more of a focus on the discussion about potential public health measures that can be done in Chile to help the issue.

Also the paper needs to explain why only population aged over 12 years were included and why not older ages. The researchers should include this in their paper. In addition please add in details how many invitations to the study have you sent and describe in details the inclusion and exclusion criteria.

Please discuss power calculation and how the sample size is adequate. This needs to be stated clearly as well. 

Author Response

Date    :                       May 19, 2021

To        :                       Ovidiu Radu, Assistant Editor, IJERPH

Re       :                       Revision of Manuscript ID: ijerph-1173675

Dear Editor,

My co-authors and I would like to thank you and the reviewers for your expert and detailed reviews and note that both reviewers and the Editorial team saw merit in our paper. We have carefully revised our manuscript, greatly benefitting from suggestions.

We have responded to each critique individually on the pages that follow (reviewer critiques in normal print and our responses in italics). Additionally, we have updated the numbers of migrants in the country considering the latest available data.

I am hopeful that this revision will be satisfactory for both reviewers and the editorial team. I look forward to hearing from you soon.

Sincerely,

Daiana Quintiliano Scarpelli Dourado, MsC, PhD.

#Reviewer 1

Comment 1: Thank you for the opportunity to review this article. In my opinion it is a very well-written article.  I appreciate the researcher’s effort and novelty of this study. I have only minor comments:  This paper analyzes the associations between lifestyle habits and health indicators in migrants and native children in Chile. Since this is a public healthy journal, I want to see more of a focus on the discussion about potential public health measures that can be done in Chile to help the issue.

Answer: We thank the reviewer for this comment. We do appreciate it! We have few national information related to this. We have cited two documents that present political regulations regarding the migrant population in Chile (Ministerio de Educación, 2018; MINSAL, 2018)(see lines 316-323). Furthermore, we mention the importance of moving towards an intercultural educational environment. So that both natives and migrants can exchange knowledge related to healthy habits. In fact, in our country much of this phenomenon stills needs to be understood, so this study takes on its importance.

Comment 2: Also, the paper needs to explain why only population aged over 12 years were included and why not older ages. The researchers should include this in their paper.

Answer: We have added more detail about the characteristics of our sample. We have made sure that it is clear in the text that the sample is made up of schoolchildren between 7 and 11.9 years of age with a median (IR) of 9.1(8.2-9.9) years old.

Our study corresponds to a secondary analysis of a research project whose methodology includes schoolchildren from second, third and fourth grades of primary education in Chile. In general, the age of schoolchildren in these grades, ranges from 7-8 years to 9-10 years of age. Children older than 12 years attending those grades are exceptions. Moreover, our study did not include data on sexual maturation (Tanner stages, for example). Thus, it was impossible to remove the effect of sexual maturation on variables such as strength or self-esteem. This information is presented in the Material and Methods section (see lines100-101).

Comment 3: In addition, please add in details how many invitations to the study have you sent and describe in details the inclusion and exclusion criteria.

Answer: We thank the reviewer for the opportunity to explain a little more about our study. Our study corresponds to a secondary analysis of a research project called “Development, scaling up and validation of an integrated system of interventions in schoolchildren in nutrition, physical activity and community environment". Initially, the sample size was calculated in 1936 schoolchildren which was representative of all students enrolled in primary schools from the southern area of Santiago. However, the initial sample was not possible to be obtained due to a sociopolitical situation that made impossible our presence in the schools. Due to this, the sample reached in this study (n=1033) although interesting, was no longer representative of the population of schoolchildren in the territory. Inclusion and exclusion criteria are detailed in lines 95-98.

Comment 4: Please discuss power calculation and how the sample size is adequate. This needs to be stated clearly as well. 

Answer: As mentioned above, our sample was obtained by convenience (non-probabilistic) and correspond to a secondary analysis of another study which was not designed for the purpose of our study. This has been mentioned in the limitations of our study (see lines 324-328). Despite such limitation, in our results we observed associations between nationality and indicators of health and life habits. We believe that the results of this study may awaken interest for the development of future studies where the limitations that we faced can be addressed.

#Reviewer 2

Comment 1: First, my acknowledgement to the editor for permit the opportunity of review this manuscript. While the manuscript and field could be of major interest, I have serious concerns. While the aim describe for the authors was "to explore the associations between lifestyle habits and health indicators in migrants and native children in Chile, in 6 municipalities of the Metropolitan Region", the authors mention along the manuscript and even in the methods section: "It is important to mention that our interest is the difference between migrant and native Chilean children, so this was the focus of our analysis." This is a serious inconsistency.  For example, table 2 provide a comparison of descriptive characteristic of sample according to nationality, and this is not an aim. 

Answer: We thank the reviewer for this careful reflection and the possibility of correction. In fact, we have noticed that the way the title, objective and results were presented was confusing. We want to clarify that our interest was to explore the associations between nationality (Chilean and migrant children) with health indicators and lifestyle habits among schoolchildren in Chile. Considering that the title and the objective are closely connected, both have been reformulated in the abstract (see lines 17-18) and Introduction (see lines 78-80).

Again, we apologize that our idea was not communicated in the best way, the comment has been of great benefit. We think that with these modifications is possible to understand our study approaches.

Comment 2: My serious concerns are about statistical comparison in two groups with quite differences in number of participants. Additionally, the results are separated by sex, without rationale. The most interesting data are in migrant population, but also are the minority in the sample and probably not totally representative. 

Answer: We thank the reviewer’s comments.  Each of the mentioned points is addressed below:

  1. Regarding comparisons between groups with different numbers of participants: We understand the reviewer’s concern when verifying that the groups (Chilean and migrant children) are dissimilar in number (839 vs 140), this also was a discussion in the team during the manuscript preparation. We would like to present the main reasons that supported our decision. Firstly, despite this is a non-probabilistic sample the proportion of migrant children in our sample is higher than the national one (13.4% vs 4.4%). Secondly, initially we tested the variances between the groups (migrant vs Chileans) in the continuous variables. All variables (handgrip strength, standing long jump, z-BMI, body fat, self-esteem and physical activity score) presented equal variance (for two-tailed variance ratio test in Stata); decreasing the probabilities of type 1 error and loss of statistical power. Thus, the comparisons between the groups is statistically acceptable and they were carried out.

  1. Regarding the results separated by sex: We would like to clarify that all the results are presented for the complete sample (both sexes) and according to sex (Table 2, 3 and Figure 1). The rationale for the analysis according to sex was based on a previous review of the literature, which several studies report different responses between boys and girls. For instance, differences according to sex for nutritional status (Mora et al., 2012), muscular fitness (Pardo-Arquero, Jimenez-Pavón, Guillén el Castillo, & Benítez-Sillero, 2014) and physical activity (Reimers et al., 2019). Analysis by sex can help identify subgroups of with higher health risk or harmful lifestyle habits. In our case, this stratification allowed us to identify differences in some of the variables, such as dietary habits and handgrip strength. Furthermore, this would allow, for example, an adaptation in interventions with a gender perspective.

  1. Regarding the representativeness of the population of migrant schoolchildren: Finally, our study does not pretend to be representative of the total population of migrant schoolchildren in Chile. Regarding this concern, additional information was included in limitations sections (see lines 324-328). Even with these limitations, we do believe that our study, being one of the first to explore the association between nationality and indicators of health and lifestyle, marks a starting point and this issue must continue being studied in Chile and in South America.

Comment 3: Statistical models are not extensively explained in method section and are not completely justified. Why the authors used time of residence in Chile, PAQ-C final score, and Z-BMI/A, regardless of their level of significance, as covariates? For example, an important variable mentioned in introduction section highly correlated with several health outcomes is vulnerability index, why was not included? Why univariate models?

Answer: We thank the reviewer for the comments and each of them is addressed below:

  1. Regarding the justification of statistical models: We appreciate the comment, we have added more detail in the methodology section that justified the analyzes carried out (see lines 155-157). As the reviewer mentions, we used time of residence in Chile, PAQ-C and Z-BMI as covariates regardless of their level of significance. This decision was based on the available evidence, where an association between the aforementioned variables and the outcomes of our study (overweight/obesity, abdominal obesity, muscular fitness, self-esteem and eating habits) has been reported. In Figure 3, we showed 3 models. Model 1 is not adjusted. Model 2, where all variables were adjusted by PAQ-C score. This due to the physical activity has been suggested correlates with nutritional status, muscular fitness and self-esteem (Kotz, Perez-Leighton, Teske, & Billington, 2017; Rodriguez-Ayllon et al., 2018). Additionally, handgrip strength and standing long jump were adjusted by Z-BMI because are influenced by body weight (handgrip) and height (standing long jump) which are the component of Z-BMI. Finally, self-esteem and accomplishment of guidelines are the only variables adjusted by age to remove the effect of maturation, and because age had already been considered in the construction of the other outcomes. In model 3, all variables were additionally adjusted by time of residence in Chile. This, due that as the time in the country pass, more assimilation of the culture and adaptation of the habits is expected (Hun Gamboa, Urzúa Morales, & López-Espinoza, 2020; Van Hook, Quiros, Frisco, & Fikru, 2016).

  1. Regarding the non-inclusion of the vulnerability index: As mentioned by the reviewer, socioeconomic factors correlate with various health outcomes, including those of our study. Unfortunately, we did not have access to information on the income level, economic status or the vulnerability index of the children/family in our sample. In relation to the use of vulnerability index of the school as a covariate, this was not considered because the indexes were very similar among the schools (mean 91.5±5.4; while the minimum was 88.0 and maximum 98.0). It is worth noting that we use the vulnerability index of the school and not of the children. The aim of mentioning it in the introduction was to characterize the context. Information about the mean, standard deviation, minimum and maximum of this vulnerability index was added in Material and Methods section (see line 93).

  1. Regarding the univariate model: According to the models presented in Figure 1, we have chosen to mention that model 1 is not adjusted, while models 2 and 3 are adjusted. This, instead of referring to its univariable nature (model 1) or multivariable (models 2 and 3). This information was modified in lines 197 and 220.

Comment 4: "Our findings suggest that migrant children in Chile (in this sample) presented better nutritional status, muscular fitness, and dietary habits than native ones." This sentence is very strong, while the data did not support completely this affirmation.

Answer: We acknowledge that this declaration we mention at the beginning of the discussion was only partially supported by our results. The intention was to summarize our observations, in a very general way. Based on this concern, we have reformulated this sentence referring exactly to what was observed in our results.

Thus, the phrase in the original version

Our findings suggest that migrant children in Chile presented better nutritional status, muscular fitness, and dietary habits than native ones.”

Have been changed by:

In our sample migrant schoolchildren presented better health indicators and lifestyle habits than the native ones, except for self-esteem that was similar in the groups. Some of these differences were possible to evidence just with the sex stratification (risk of low handgrip strength, for boys; and risk of unsatisfactory/partial food guidelines accomplishment, for girls).” (See lines 226-229).

Comment 5: I suggest to the authors consider including socioeconomic factors in the statistical models. These variables may be confounding seriously the results in a profoundly inequially Country as Chile.

Answer: We wish we had done that, unfortunately we did not have access to the personal socioeconomic information from the families. The available data is just those provided by the schools and municipalities in the public databases. 

References

Hun Gamboa, N., Urzúa Morales, A., & López-Espinoza, A. (2020). Food and migration: a descriptive-comparative analysis of food behavior between Chileans and Colombians residing in the north and center of Chile. Nutrición Hospitalaria, 37(4), 823–829. https://doi.org/10.20960/nh.03035

Kotz, C. M., Perez-Leighton, C. E., Teske, J. A., & Billington, C. J. (2017). Spontaneous Physical Activity Defends Against Obesity. Current Obesity Reports, 6(4), 362–370. https://doi.org/10.1007/s13679-017-0288-1

Ministerio de Educación. (2018). Política Nacional de Estudiantes Extranjeros 2018-2022. Retrieved from https://migrantes.mineduc.cl/wp-content/uploads/sites/88/2020/04/Política-Nacional-Estud-Extranjeros.pdf

MINSAL. (2018). Política De Salud De Migrantes Internacionales. Politica De Salud De Migrantes Internacionales, 1–57. Retrieved from http://redsalud.ssmso.cl/wp-content/uploads/2018/01/Politica-de-Salud-de-Migrantes-310-1750.pdf

Mora, A. I., Lopez-Ejeda, N., Anzid, K., Montero, P., Marrodan, M. D., & Cherkaoui, M. (2012). Influencia de la migración en el estado nutricional y comportamiento alimentario de adolescentes marroquíes residentes en Madrid (España). Nutricion Clinica y Dietetica Hospitalaria, 32(SUPPL.2), 48–54.

Pardo-Arquero, V., Jimenez-Pavón, D., Guillén el Castillo, M., & Benítez-Sillero, J. (2014). Physical activity, fitness and adiposity: inmigrants versus spanish scholars. Rev Internacional de Medicina y Ciencias de La Actividad Física y El Deporte, 14(54), 319–338.

Reimers, A. K., Brzoska, P., Niessner, C., Schmidt, S. C. E., Worth, A., & Woll, A. (2019). Are there disparities in different domains of physical activity between school-aged migrant and non-migrant children and adolescents? Insights from Germany. PLoS ONE, 14(3), 1–14. https://doi.org/10.1371/journal.pone.0214022

Rodriguez-Ayllon, M., Cadenas-Sanchez, C., Esteban-Cornejo, I., Migueles, J. H., Mora-Gonzalez, J., Henriksson, P., … Ortega, F. B. (2018). Physical fitness and psychological health in overweight/obese children: A cross-sectional study from the ActiveBrains project. Journal of Science and Medicine in Sport, 21(2), 179–184. https://doi.org/10.1016/j.jsams.2017.09.019

Van Hook, J., Quiros, S., Frisco, M. L., & Fikru, E. (2016). It is Hard to Swim Upstream: Dietary Acculturation Among Mexican-Origin Children. Population Research and Policy Review, 35(2), 177–196. https://doi.org/10.1007/s11113-015-9381-x

Reviewer 2 Report

First, my acknowledgement to te editor for permit  the opportunity of review this manuscript. 

While the manuscript and field could be of major interest, I have serious concerns.

While the aim describe for the authors was " to explore the associations between lifestyle habits and health indicators in migrants and native children in Chile, in 6 municipalities of the Metropolitan Region", the authors mention along the manuscript and even in the methods section: "It is important to mention that our interest is the difference between migrant and native Chilean children, so this was the focus of our analysis." This is a serious inconsistency.  For example, table 2 provide a comparison of descriptive characteristic of sample according to nationallity, and this is not an aim. 

My serious concerns is about statistical comparison in two groups with quite differences in number of participants. Additionally, the results are separated by sex, without rationale. The most interesintg data are in migrant population, but also are the minority in the sample and probably not totally representative. 

Statistical models are not extensively explained in method section, and are not completely justified. Why the authors used  time of residence in Chile, PAQ-C final score, and Z-BMI/A, regardless of their level of significance, as covariates? For example, an important variable mentioned in introducction section hilghy correlated with several health outcomes is vulnerability index, why was not included? Why univariate models?

"Our findings suggest that migrant children in Chile presented better nutritional status, muscular fitness, and dietary habits than native ones." This sentence is very strong, while the data did not support completelly this affirmation.

I suggest to the authors consider including socioeconimic factors in the statistical models. These variables may be confounding seriouly the results in a profondly inequially Country as Chile.

Author Response

(The authors gave the same response as above.)

Round 2

Reviewer 2 Report

The manuscript is improved and the authors have addressed my concerns. Congrats.